# Metric on Nonlinear Dynamical Systems
# with Perron-Frobenius Operators

**Isao Ishikawa**[†‡]**, Keisuke Fujii**[†]**, Masahiro Ikeda**[†‡]**, Yuka Hashimoto**[†‡]**, Yoshinobu Kawahara**[†§]
[†]RIKEN Center for Advanced Intelligence Project
[‡]School of Fundamental Science and Technology, Keio University
[§]The Institute of Scientific and Industrial Research, Osaka University
{isao.ishikawa, keisuke.fujii.zh, masahiro.ikeda}@riken.jp
yukahashimoto@keio.jp, ykawahara@sanken.osaka-u.ac.jp

## Abstract

The development of a metric for structural data is a long-term problem in pattern recognition and machine learning. In this paper, we develop a general metric for comparing nonlinear dynamical systems that is defined with Perron-Frobenius operators in reproducing kernel Hilbert spaces. Our metric includes the existing fundamental metrics for dynamical systems, which are basically defined with principal angles between some appropriately-chosen subspaces, as its special cases. We also describe the estimation of our metric from finite data. We empirically illustrate our metric with an example of rotation dynamics in a unit disk in a complex plane, and evaluate the performance with real-world time-series data.

## 1 Introduction

Classification and recognition has been one of the main focuses of research in machine learning for the past decades. When dealing with some structural data other than vector-valued ones, the development of an algorithm for this problem according to the type of the structure is basically reduced to the design of an appropriate metric or kernel. However, not much of the existing literature has addressed the design of metrics in the context of dynamical systems. To the best of our knowledge, the metric for ARMA models based on comparing their cepstrum coefficients [12] is one of the first papers to address this problem. De Cock and De Moor extended this to linear state-space models by considering the subspace angles between the observability subspaces [6]. Meanwhile, Vishwanathan et al. developed a family of kernels for dynamical systems based on the Binet-Cauchy theorem [25]. Chaudhry and Vidal extended this to incorporate the invariance on initial conditions [4].

As mentioned in some of the above literature, the existing metrics for dynamical systems that have been developed are defined with principal angles between some appropriate subspaces such as column subspaces of observability matrices. However, those are basically restricted to linear dynamical systems although Vishwanathan et al. mentioned an extension with reproducing kernels for some specific metrics [25]. Recently, Fujii et al. discussed a more general extension of these metrics to nonlinear systems with Koopman operator [8]. Mezic et al. propose metrics of dynamcal systems in the context of ergodic theory via Koopman operators on $L^2$-spaces[14, 15]. The Koopman operator, also known as the composition operator, is a linear operator on an observable for a nonlinear dynamical system [10]. Thus, by analyzing the operator in place of directly nonlinear dynamics, one could extract more easily some properties about the dynamics. In particular, spectral analysis of Koopman operator has attracted attention with its empirical procedure called dynamic mode decomposition (DMD) in a variety of fields of science and engineering [18, 2, 17, 3].

In this paper, we develop a general metric for nonlinear dynamical systems, which includes the existing fundamental metrics for dynamical systems mentioned above as its special cases. This metric

is defined with Perron-Frobenius operators in reproducing kernel Hilbert spaces (RKHSs), which are shown to be essentially equivalent to Koopman operators, and allows us to compare a pair of datasets that are supposed to be generated from nonlinear systems. We also describe the estimation of our metric from finite data. We empirically illustrate our metric using an example of rotation dynamics in a unit disk in a complex plane, and evaluate the performance with real-world time-series data.

The remainder of this paper is organized as follows. In Section 2, we first briefly review the definition of Koopman operator, especially the one defined in RKHSs. In Section 3, we give the definition of our metric for comparing nonlinear dynamical systems (NLDSs) with Koopman operators and, then, describe the estimation of the metric from finite data. In Section 4, we describe the relation of our metric to the existing ones. In Section 5, we empirically illustrate our metric with synthetic data and evaluate the performance with real-world data. Finally, we conclude this paper in Section 6.

## 2 Perron-Frobenius operator in RKHS

Consider a discrete-time nonlinear dynamical system $\boldsymbol{x}_{t+1} = \boldsymbol{f}(\boldsymbol{x}_t)$ with time index $t \in \mathbb{T} := \{0\} \cup \mathbb{N}$ and defined on a state space $\mathcal{M}$ (i.e., $\boldsymbol{x} \in \mathcal{M}$), where $\boldsymbol{x}$ is the state vector and $\boldsymbol{f} \colon \mathcal{M} \to \mathcal{M}$ is a (possibly, nonlinear) state-transition function. Then, *the Koopman operator* (also known as the composition operator), which is denoted by $\mathcal{K}$ here, is a linear operator in a function space $X$ defined by the rule

$$\mathcal{K}g = g \circ \boldsymbol{f}, \tag{1}$$

where $g$ is an element of $X$. The domain $\mathscr{D}(\mathcal{K})$ of the Koopman operator $\mathcal{K}$ is $\mathscr{D}(\mathcal{K}) := \{g \in X \mid g \circ \boldsymbol{f} \in X\}$, where $\circ$ denotes the composition of $g$ with $\boldsymbol{f}$ [10]. The choice of $X$ depends on the problem considered. In this paper, we consider $X$ as an RKHS. The function $g$ is referred as *an observable*. We see that $\mathcal{K}$ acts linearly on the function $g$, even though the dynamics defined by $\boldsymbol{f}$ may be nonlinear. In recent years, spectral decomposition methods for this operator has attracted attention in a variety of scientific and engineering fields because it could give a global modal description of a nonlinear dynamical system from data. In particular, a variant of estimation algorithms, called dynamic mode decomposition (DMD), has been successfully applied in many real-world problems, such as image processing [11], neuroscience [3], and system control [16]. In the community of machine learning, several algorithmic improvements have been investigated by a formulation with reproducing kernels [9] and in a Bayesian framework [22].

Now, let $\mathcal{H}_k$ be the RKHS endowed with a dot product $\langle \cdot, \cdot \rangle$ and a positive definite kernel $k \colon \mathcal{X} \times \mathcal{X} \to \mathbb{C}$ (or $\mathbb{R}$), where $\mathcal{X}$ is a set. Here, $\mathcal{H}_k$ is a function space on $\mathcal{X}$. The corresponding feature map is denoted by $\phi \colon \mathcal{X} \to \mathcal{H}_k$. Also, assume $\mathcal{M} \subset \mathcal{X}$, and define the closed subspace $\mathcal{H}_{k,\mathcal{M}} \subset \mathcal{H}_k$ by the closure of the vector space generated by $\phi(\boldsymbol{x})$ for $\forall \boldsymbol{x} \in \mathcal{M}$, i.e. $\mathcal{H}_{k,\mathcal{M}} := \overline{\operatorname{span}\{\phi(\boldsymbol{x}) \mid \boldsymbol{x} \in \mathcal{M}\}}$. Then, the *Perron-Frobenius operator in RKHS* associated with $\boldsymbol{f}$ (see [9], note that $K_f$ is called Koopman operator on the feature map $\phi$ in the literature), $K_{\boldsymbol{f}} \colon \mathcal{H}_{k,\mathcal{M}} \to \mathcal{H}_{k,\mathcal{M}}$, is defined as a linear operator with dense domain $\mathscr{D}(K_f) := \operatorname{span}(\phi(\mathcal{M}))$ satisfying for all $\boldsymbol{x} \in \mathcal{M}$,

$$K_{\boldsymbol{f}}[\phi(\boldsymbol{x})] = \phi(\boldsymbol{f}(\boldsymbol{x})). \tag{2}$$

Since $K_f$ is densely defined, there exists the adjoint operator $K_f^*$. In the following proposition, we see that $K_f^*$ is essentially the same as Koopman operator $\mathcal{K}$.

**Proposition 2.1.** *Let $X = H$ be the RKHS associated with the positive definite kernel $k|_{\mathcal{M} \times \mathcal{M}}$ defined by the restriction of $k$ to $\mathcal{M} \times \mathcal{M}$, which is a function space on $\mathcal{M}$. Let $\rho \colon \mathcal{H}_{k,\mathcal{M}} \to H$ be a linear isomorphism defined via the restriction of functions from $\mathcal{X}$ to $\mathcal{M}$. Then, we have*

$$\rho K_{\boldsymbol{f}}^* \rho^{-1} = \mathcal{K},$$

*where $(\cdot)^*$ means the Hermitian transpose.*

*Proof.* Let $g \in \mathscr{D}(\mathcal{K})$. Since the feature map for $H$ is the same as $\rho \circ \phi$, by the reproducing property, $\langle g, \rho K_f(\phi(\boldsymbol{x})) \rangle_H = \langle g, \rho \circ \phi(\boldsymbol{f}(\boldsymbol{x})) \rangle_H = g \circ \boldsymbol{f}(\boldsymbol{x}) = \langle \mathcal{K}g, \rho \circ \phi(\boldsymbol{x}) \rangle_H$. Thus the definitions (1), (2), and the fact $\rho^* = \rho^{-1}$ show the statement. □

## 3 Metric on NLDSs with Perron-Frobenius Operators in RKHSs

We propose a general metric for the comparison of nonlinear dynamical systems, which is defined with Perron-Frobenius operators in RKHSs. Intuitively, the metric compares the behaviors of dynamical

systems over infinite time. To ensure the convergence property, we consider the ratio of metrics, namely angles instead of directly considering exponential decay terms. We first give the definition in Subsection 3.1, and then derive an estimator of the metric from finite data in Subsection 3.2.

## 3.1 Definition

Let $\mathcal{H}_{\mathrm{ob}}$ be a Hilbert space and $\mathcal{M} \subset \mathcal{X}$ a subset. Let $h : \mathcal{M} \to \mathcal{H}_{\mathrm{ob}}$ be a map, often called an observable. We define the *observable operator* for $h$ by a linear operator $L_h : \mathcal{H}_{k,\mathcal{M}} \to \mathcal{H}_{\mathrm{ob}}$ such that $h = L_h \circ \phi$. We give two examples here: First, in the case of $\mathcal{H}_{\mathrm{ob}} = \mathbb{C}^d$ and $h(\boldsymbol{x}) = (g_1(\boldsymbol{x}), \ldots, g_m(\boldsymbol{x}))$ for some $g_1, \ldots, g_m \in \mathcal{H}_k$, the observable operator is $L_h(v) := (\langle g_i, v \rangle)_{i=1}^m$. This situation appears, for example, in the context of DMD, where observed data is obtained by values of functions in RKHS. Secondly, in the case of $\mathcal{H}_{\mathrm{ob}} = \mathcal{H}_{k,\mathcal{M}}$ and $h = \phi|_{\mathcal{M}}$, the observable operator is $L_h(v) = v$. This situation appears when we can observe the state space $\mathcal{X}$, and we try to get more detailed information by observing data sent to RKHS via the feature map.

Let $\mathcal{H}_{\mathrm{in}}$ be a Hilbert space. we refer to $\mathcal{H}_{\mathrm{in}}$ as an *initial value space*. We call a linear operator $\mathscr{I} : \mathcal{H}_{\mathrm{in}} \to \mathcal{H}_{k,\mathcal{M}}$ *an initial value operator* on $\mathcal{M}$ if $\mathscr{I}$ is a bounded operator. Initial value operators are regarded as expressions of initial values in terms of linear operators. In fact, in the case of $\mathcal{H}_{\mathrm{in}} = \mathbb{C}^N$ and let $\boldsymbol{x}_1, \ldots, \boldsymbol{x}_N \in \mathcal{M}$. Let $\mathscr{I} := (\phi(\boldsymbol{x}_1), \ldots, \phi(\boldsymbol{x}_N))$ be an initial value operator on $\mathcal{M}$, which is a linear operator defined by $\mathscr{I}((a_i)_{i=1}^N) = \sum_i a_i \phi(\boldsymbol{x}_i)$. Let $K_{\boldsymbol{f}}$ be a Perron-Frobenius operator associated with a dynamical system $\boldsymbol{f} : \mathcal{M} \to \mathcal{M}$. Then for any positive integer $n > 0$, we have $K_{\boldsymbol{f}}^n \mathscr{I}((a_i)_{i=1}^N) = \sum_i a_i \phi(\boldsymbol{f}^n(\boldsymbol{x}_i))$, and $K_{\boldsymbol{f}}^n \mathscr{I}$ is a linear operator including information at time $n$ of the orbits of the dynamical system $\boldsymbol{f}$ with inital values $\boldsymbol{x}_1, \ldots, \boldsymbol{x}_N$.

Now, we define *triples of dynamical systems*. A triple of a dynamical system with respect to an initial value space $\mathcal{H}_{\mathrm{in}}$ and an observable space $\mathcal{H}_{\mathrm{ob}}$ is a triple $(\boldsymbol{f}, h, \mathscr{I})$, where the first component $\boldsymbol{f} : \mathcal{M} \to \mathcal{M}$ is a dynamical system on a subset $\mathcal{M} \subset \mathcal{X}$ ($\mathcal{M}$ depends on $\boldsymbol{f}$) with Perron-Frobenius operator $K_{\boldsymbol{f}}$, the second component $h : \mathcal{M} \to \mathcal{H}_{\mathrm{ob}}$ is an observable with an observable operator $L_h$, and the third component $\mathscr{I} : \mathcal{H}_{\mathrm{in}} \to \mathcal{H}_{k,\mathcal{M}}$ is an initial value operator on $\mathcal{M}$, such that for any $r \geq 0$, the composition $L_h K_{\boldsymbol{f}}^r \mathscr{I}$ is well-defined and a Hilbert Schmidt operator. We denote by $\mathscr{T}(\mathcal{H}_{\mathrm{in}}, \mathcal{H}_{\mathrm{ob}})$ the set of triples of dynamical systems with respect to an initial value space $\mathcal{H}_{\mathrm{in}}$ and an observable space $\mathcal{H}_{\mathrm{ob}}$.

For two triples $D_1 = (\boldsymbol{f}_1, h_1, \mathscr{I}_1), D_2 = (\boldsymbol{f}_2, h_2, \mathscr{I}_2) \in \mathscr{T}(\mathcal{H}_{\mathrm{in}}, \mathcal{H}_{\mathrm{ob}})$, and for $T, m \in \mathbb{N}$, we first define

$$\mathfrak{K}_m^T (D_1, D_2) := \mathrm{tr} \left( \bigwedge^m \sum_{r=0}^{T-1} \left( L_{h_2} K_{\boldsymbol{f}_2}^r \mathscr{I}_2 \right)^* L_{h_1} K_{\boldsymbol{f}_1}^r \mathscr{I}_1 \right) \in \mathbb{C},$$

where the symbol $\wedge^m$ is the $m$-th exterior product (see Appendix A). We note that since $K_{\boldsymbol{f}_i}$ is bounded, we regard $K_{\boldsymbol{f}_i}$ as a unique extension of $K_{\boldsymbol{f}_i}$ to a bounded linear operator with domain $\mathcal{H}_{k,\mathcal{M}}$.

**Proposition 3.1.** *The function $\mathfrak{K}_m^T$ is a positive definite kernel on $\mathscr{T}(\mathcal{H}_{\mathrm{in}}, \mathcal{H}_{\mathrm{ob}})$.*

*Proof.* See Appendix B. $\qquad\square$

Next, for positive number $\varepsilon > 0$, we define $A_m^T$ with $\mathfrak{K}_m^T$ by

$$A_m^T (D_1, D_2) := \lim_{\epsilon \to +0} \frac{\left| \epsilon + \mathfrak{K}_m^T (D_1, D_2) \right|^2}{(\epsilon + \mathfrak{K}_m^T (D_1, D_1)) (\epsilon + \mathfrak{K}_m^T (D_2, D_2))} \in [0, 1].$$

We remark that for $D \in \mathscr{T}(\mathcal{H}_{\mathrm{in}}, \mathcal{H}_{\mathrm{ob}})$, $\left( \mathfrak{K}_m^T(D, D) \right)_{T=1}^\infty$ is a non-negative increasing sequence. Now, we denote by $\ell^\infty$ the Banach space of bounded sequences of complex numbers, and define $\mathbf{A}_m : \mathscr{T}(\mathcal{H}_{\mathrm{in}}, \mathcal{H}_{\mathrm{ob}})^2 \to \ell^\infty$ by

$$\mathbf{A}_m := \left( A_m^T \right)_{T=1}^\infty$$

Moreover, we introduce *Banach limits* for elements of $\ell^\infty$. The Banach limit is a bounded linear functional $\mathcal{B} : \ell^\infty \to \mathbb{C}$ satisfying $\mathcal{B}((1)_{n=1}^\infty) = 1$, $\mathcal{B}((z_n)_{n=1}^\infty) = \mathcal{B}((z_{n+1})_{n=1}^\infty)$ for any $(z_n)_n$, and $\mathcal{B}((z_n)_{n=1}^\infty) \geq 0$ for any non-negative real sequence $(z_n)_{n=1}^\infty$, namely $z_n \geq 0$ for all $n \geq 1$. We remark that if $(z_n)_n \in \ell^\infty$ converges a complex number $\alpha$, then for any Banach limit $\mathcal{B}$, $\mathcal{B}((z_n)_{n=1}^\infty) = \alpha$. The existence of the Banach limits is first introduced by Banach [1] and proved through the Hahn-Banach theorem. In general, the Banach limit is not unique.

**Definition 3.1.** For an integer $m > 0$ and a Banach limit $\mathcal{B}$, a positive definite kernel $\mathscr{A}_m^{\mathcal{B}}$ is defined by

$$\mathscr{A}_m^{\mathcal{B}} := \mathcal{B}\left(\mathbf{A}_m\right).$$

We remark that positive definiteness of $\mathscr{A}_m^{\mathcal{B}}$ follows Proposition 3.1 and the properties of the Banach limit. We then simply denote $\mathscr{A}_m^{\mathcal{B}}(D_1, D_2)$ by $\mathscr{A}_m(D_1, D_2)$ if $\mathbf{A}_m(D_1, D_2)$ converges since that is independent of the choice of $\mathcal{B}$.

In general, a Banach limit $\mathcal{B}$ is hard to compute. However, under some assumption and suitable choice of $\mathcal{B}$, we prove that $\mathscr{A}_m^{\mathcal{B}}$ is computable in Proposition 3.6 below. Thus, we obtain an estimation formula of $\mathscr{A}_m^{\mathcal{B}}$ (see [20], [21], [7] for other results on the estimation of Banach limit). In the following proposition, we show that we can construct a pseudo-metric from the positive definite kernel $\mathscr{A}_m^{\mathcal{B}}$:

**Proposition 3.2.** *Let $\mathcal{B}$ be a Banach limit. For $m > 0$, $\sqrt{1 - \mathscr{A}_m^{\mathcal{B}}(\cdot, \cdot)}$ is a pseudo-metric on $\mathscr{T}(\mathcal{H}_{\mathrm{in}}, \mathcal{H}_{\mathrm{ob}})$.*

*Proof.* See Appendix C. □

**Remark 3.3.** Although we defined $\mathfrak{K}_m^T$ with RKHS, it can be defined in a more general situation as follows. Let $\mathcal{H}$, $\mathcal{H}'$ and $\mathcal{H}''$ be Hilbert spaces. For $i = 1, 2$, let $V_i \subset \mathcal{H}$ be a closed subspace, $K_i \colon V_i \to V_i$ and $L_i \colon V_i \to \mathcal{H}''$ linear operators, and let $\mathscr{I}_i \colon \mathcal{H}' \to V_i$ be a bounded operator. Then, we can define $\mathfrak{K}_m^T$ between the triples $(K_1, L_1, \mathscr{I}_1)$ and $(K_2, L_2, \mathscr{I}_2)$ in the similar manner.

### 3.2 Estimation from finite data

Now we derive an formula to compute the above metric from finite data, which allows us to compare several time-series data generated from dynamical systems just by evaluating the values of kernel functions. First, we argue the computability of $\mathscr{A}_m^{\mathcal{B}}(D_1, D_2)$ and then state the formula for computation. In this section, the initial value space is of finite dimension: $\mathcal{H}_{\mathrm{in}} = \mathbb{C}^N$, and for $v_1, \ldots, v_N \in \mathcal{H}_{k, \mathcal{M}}$. We define a linear operator $(v_1, \ldots, v_N) : \mathbb{C}^N \to \mathcal{H}_{k, \mathcal{M}}$ by $(a_i)_{i=1}^N \mapsto \sum_{i=1}^N a_i v_i$. We note that any linear operator $\mathscr{I} : \mathcal{H}_{\mathrm{in}} = \mathbb{C}^N \to \mathcal{H}_{k, \mathcal{M}}$ is an initial value operator), and, by putting $v_i := \mathscr{I}((0, \ldots, 0, \overset{i}{1}, 0, \ldots, 0))$, we have $\mathscr{I} = (v_1, \ldots, v_N)$.

**Definition 3.4.** *Let $D = (\boldsymbol{f}, h, \mathscr{I}) \in \mathscr{T}(\mathbb{C}^N, \mathcal{H}_{\mathrm{ob}})$. We call $D$ admissible if there exists $K_{\boldsymbol{f}}$'s eigen-vectors $\varphi_1, \varphi_2, \cdots \in \mathcal{H}_{k, \mathcal{M}}$ with $\|\varphi_n\| = 1$ and $K_{\boldsymbol{f}} \varphi_n = \lambda_n \varphi_n$ for all $n \geq 0$ such that $|\lambda_1| \geq |\lambda_2| \geq \ldots$ and each $v_i$ is expressed as $v_i = \sum_{n=1}^\infty a_{i,n} \varphi_n$ with $\sum_{n=1}^\infty |a_{i,n}| < \infty$, where $v_i := \mathscr{I}((0, \ldots, 0, \overset{i}{1}, 0, \ldots, 0))$.*

**Definition 3.5.** *The triple $D = (\boldsymbol{f}, h, \mathscr{I}) \in \mathscr{T}(\mathbb{C}^N, \mathcal{H}_{\mathrm{ob}})$ is semi-stable if $D$ is admissible and $|\lambda_1| \leq 1$.*

Then, we have the following asymptotic properties of $\mathbf{A}_m$.

**Proposition 3.6.** *Let $D_1, D_2 \in \mathscr{T}(\mathbb{C}^N, \mathcal{H}_{\mathrm{ob}})$. If $D_1$ and $D_2$ are semi-stable, then the sequence $\mathbf{A}_m(D_1, D_2)$ converges and the limit is equal to $\mathscr{A}_m^{\mathcal{B}}(D_1, D_2)$ for any Banach limit $\mathcal{B}$. Similarly, let $C$ be the Cesàro operator, namely, $C$ is defined to be $C((x_n)_{n=1}^\infty) := \left(n^{-1} \sum_{k=1}^n x_n\right)_{n=1}^\infty$. If $D_1$ and $D_2$ are admissible, then $C\mathbf{A}_m(D_1, D_2)$ converges and the limit is equal to $\mathscr{A}_m^{\mathcal{B}}(D_1, D_2)$ for any Banach limit $\mathcal{B}$ with $\mathcal{B}C = \mathcal{B}$.*

*Proof.* See Appendix D. □

We note that it is proved that there exists a Banach limit with $\mathcal{B}C = \mathcal{B}$ [19, Theorem 4]. The admissible or semi-stable condition holds in many cases, for example, in our illustrative example (Section 5.1).

Now, we derive an estimation formula of the above metric from finite time-series data. To this end, we first need the following lemma:

**Lemma 3.7.** *Let* $D_1 = \left( \boldsymbol{f}_1, h_1, (v_{1,l})_{l=1}^N \right), D_2 = \left( \boldsymbol{f}_2, h_2, (v_{2,l})_{l=1}^N \right) \in \mathscr{T}\left( \mathbb{C}^N, \mathcal{H}_{\mathrm{ob}} \right).$ *Then we have the following formula:*

$$
\begin{aligned}
&\mathfrak{K}_m^T(D_1, D_2) \\
&= \sum_{t_1,\dots,t_m=0}^{T-1} \sum_{\substack{0<s_1<\dots \\ <s_m\le N}} \left\langle L_{h_i} K_{\boldsymbol{f}_i}^{t_1} v_{i,s_1} \wedge \dots \wedge L_{h_i} K_{\boldsymbol{f}_i}^{t_m} v_{i,s_m}, L_{h_j} K_{\boldsymbol{f}_j}^{t_1} v_{j,s_1} \wedge \dots \wedge L_{h_j} K_{\boldsymbol{f}_j}^{t_m} v_{j,s_m} \right\rangle
\end{aligned}
$$

*Proof.* See Appendix E. $\qquad \square$

For $i = 1, 2$, we consider $N$ time-series sequences $\{y_{i,0}^l, y_{i,1}^l, y_{i,2}^l, \dots\} \subset \mathcal{H}_{\mathrm{ob}}$ in an observable space ($l = 1, \dots, N$), which are supposed to be generated from dynamical system $\boldsymbol{f}_i$ on $\mathcal{M}_i \subset \mathcal{X}$ and observed via $h_i$. That is, we consider, for $i = 1, 2, t \in \mathbb{T}$, and $l = 1, \dots, N$,

$$
\boldsymbol{x}_{i,t+1}^l = \boldsymbol{f}_i\left( \boldsymbol{x}_{i,t}^l \right), \; y_{i,t}^l = h_i(\boldsymbol{x}_{i,t}^l), \; \boldsymbol{x}_{i,0}^l \in \mathcal{M}_i. \tag{3}
$$

Assume for $i = 1, 2$, the triple $D_i = \left( \boldsymbol{f}_i, h_i, (\phi(\boldsymbol{x}_{i,0}^l))_{l=1}^N \right)$ is in $\mathscr{T}\left( \mathbb{C}^N, \mathcal{H}_{\mathrm{ob}} \right)$. Then, from Lemma 3.7, we have

$$
\begin{aligned}
&\mathfrak{K}_m^T(D_1, D_2) \\
&= \sum_{t_1,\dots,t_m=0}^{T-1} \sum_{\substack{0<s_1<\dots \\ <s_m\le N}} \left\langle L_{h_i}\phi\left( \boldsymbol{x}_{i,t_1}^{s_1} \right) \wedge \dots \wedge L_{h_i}\phi\left( \boldsymbol{x}_{i,t_m}^{s_m} \right), L_{h_j}\phi\left( \boldsymbol{x}_{j,t_1}^{s_1} \right) \wedge \dots \wedge L_{h_j}\phi\left( \boldsymbol{x}_{j,t_m}^{s_m} \right) \right\rangle \\
&= \sum_{t_1,\dots,t_m=0}^{T-1} \sum_{0<s_1<\dots<s_m\le N} \det \begin{pmatrix} \left\langle y_{i,t_1}^{s_1}, y_{j,t_1}^{s_1} \right\rangle_{\mathcal{H}_{\mathrm{ob}}} & \cdots & \left\langle y_{i,t_1}^{s_1}, y_{j,t_m}^{s_m} \right\rangle_{\mathcal{H}_{\mathrm{ob}}} \\ \vdots & \ddots & \vdots \\ \left\langle y_{i,t_m}^{s_m}, y_{j,t_1}^{s_1} \right\rangle_{\mathcal{H}_{\mathrm{ob}}} & \cdots & \left\langle y_{i,t_m}^{s_m}, y_{j,t_m}^{s_m} \right\rangle_{\mathcal{H}_{\mathrm{ob}}} \end{pmatrix}. \tag{4}
\end{aligned}
$$

In the case of $\mathcal{H}_{\mathrm{ob}} = \mathcal{H}_k$ and $h_i = \phi|_{\mathcal{M}_i}$, we see that $\left\langle y_{i,t_b}^{s_a}, y_{j,t_d}^{s_c} \right\rangle_{\mathcal{H}_{\mathrm{ob}}} = k(x_{i,t_b}^{s_a}, x_{j,t_d}^{s_c})$. Therefore, by Proposition 3.6, if $D_i$'s are semi-stable or admissible, then we can compute an convergent estimator of $\mathscr{A}_m^{\mathcal{B}}$ through $A_m^T$ just by evaluating the values of kernel functions.

## 4 Relation to Existing Metrics on Dynamical Systems

In this section, we show that our metric covers the existing metrics defined in the previous works [12, 6, 25]. That is, we describe the relation to the metric via subspace angles and Martin's metric in Subsection 4.1 and the one to the Binet-Chaucy metric for dynamical systems in Subsection 4.2 as the special cases of our metric.

### 4.1 Relation to metric via principal angles and Martin's metric

In this subsection, we show that in a certain situation, our metric reconstruct the metric (Definition 2 in [12]) for the ARMA models introduced by Martin [12] and DeCock-DeMoor [6]. Moreover, our formula generalizes their formula to the non-stable case, that is, we do not need to assume the eigenvalues are strictly smaller than 1.

We here consider two linear dynamical systems. That is, in Eqs. (3), let $\boldsymbol{f}_i \colon \mathbb{R}^q \to \mathbb{R}^q$ and $h_i \colon \mathbb{R}^q \to \mathbb{R}^r$ be linear maps for $i = 1, 2$ with $l = 1$, which we respectively denote by $\boldsymbol{A}_i$ and $\boldsymbol{C}_i$. Then, De Cock and De Moor propose to compare these two models by using the subspace angles as

$$
d((\boldsymbol{A}_1, \boldsymbol{C}_1), (\boldsymbol{A}_2, \boldsymbol{C}_2)) = -\log \prod_{i=1}^m \cos^2 \theta_i, \tag{5}
$$

where $\theta_i$ is the $i$-th subspace angle between the column spaces of the extended observability matrices $\mathcal{O}_i := [\boldsymbol{C}_i^\top \; (\boldsymbol{C}_i \boldsymbol{A}_i)^\top \; (\boldsymbol{C}_i \boldsymbol{A}_i^2)^\top \; \cdots]$ for $i = 1, 2$. Meanwhile, Martin define a distance on AR models via cepstrum coefficients, which is later shown to be equivalent to the distance (5) [6].

Now, we regard $\mathcal{X} = \mathbb{R}^q$. The positive definite kernel here is the usual inner product of $\mathbb{R}^q$ and the associated RKHS is canonically isomorphic to $\mathbb{C}^q$. Let $\mathcal{H}_{\mathrm{in}} = \mathbb{C}^q$ and $\mathcal{H}_{\mathrm{ob}} = \mathbb{C}^r$. Note that for

$i = 1, 2$, $\boldsymbol{D}_i = (\boldsymbol{A}_i, \boldsymbol{C}_i, \boldsymbol{I}_q) \in \mathscr{T}(\mathbb{C}^q, \mathbb{C}^r)$, and for any linear maps $\boldsymbol{f} : \mathbb{R}^q \to \mathbb{R}^q$ and $\boldsymbol{h} : \mathbb{R}^q \to \mathbb{R}^N$, $K_{\boldsymbol{f}} = \boldsymbol{f}$ and $L_{\boldsymbol{h}} = \boldsymbol{h}$.

Then we have the following theorem:

**Proposition 4.1.** *The sequence* $\mathbf{A}_q(\boldsymbol{D}_1, \boldsymbol{D}_2)$ *converges. In the case that the systems are observable and stable, this limit* $\mathscr{A}_q(\boldsymbol{D}_1, \boldsymbol{D}_2)$ *is essentially equal to* (5).

*Proof.* See Appendix F. $\qquad\square$

Therefore, we can define a metric between linear dynamical systems with $(\boldsymbol{A}_1, \boldsymbol{C}_1)$ and $(\boldsymbol{A}_2, \boldsymbol{C}_2)$ by $\mathscr{A}_q(\boldsymbol{D}_1, \boldsymbol{D}_2)$.

Moreover, the value $\mathscr{A}_q(\boldsymbol{D}_1, \boldsymbol{D}_2)$ captures an important characteristic of behavior of dynamical systems. We here illustrate it in the situation where the state space models come from AR models. We will see that $\mathscr{A}_q(\boldsymbol{D}_1, \boldsymbol{D}_2)$ has a sensitive behavior on the unit circle, and gives a reasonable generalization of Martin's metric [12] to the non-stable case.

For $i = 1, 2$, we consider an observable AR model:

$$(M_i) \quad \boldsymbol{y}_t = a_{i,1}\boldsymbol{y}_{t-1} + \cdots + a_{i,q}\boldsymbol{y}_{t-q}, \tag{6}$$

where $a_{i,k} \in \mathbb{R}$ for $k \in \{1, \cdots, q\}$. Let $\boldsymbol{C}_i = (1, 0, \ldots, 0) \in \mathbb{C}^{1 \times q}$, and let $\boldsymbol{A}_i$ be the companion matrix for $M_i$. And, let $\gamma_{i,1}, \ldots, \gamma_{i,q}$ be the roots of the equation $y^q - a_{i,1}y^{q-1} - \cdots - a_{i,q} = 0$. For simplicity, we assume these roots are distinct complex numbers. Then, we define

$$P_i := \left\{ \gamma_{i,n} \mid |\gamma_{i,n}| > 1 \right\}, \; Q_i := \left\{ \gamma_{i,n} \mid |\gamma_{i,n}| = 1 \right\}, \text{and} \; R_i := \left\{ \gamma_{i,n} \mid |\gamma_{i,n}| < 1 \right\}.$$

As a result, if $|P_1| = |P_2|$, $|R_1| = |R_2|$, and $Q_1 = Q_2$, we have

$$\mathscr{A}_q(\boldsymbol{D}_1, \boldsymbol{D}_2)$$
$$= \frac{\displaystyle\prod_{\alpha,\beta \in P_1}\left(1 - \alpha\overline{\beta}\right) \cdot \prod_{\alpha,\beta \in P_2}\left(1 - \alpha\overline{\beta}\right)}{\displaystyle\prod_{\alpha \in P_1, \beta \in P_2}|1 - \alpha\beta|^2} \cdot \frac{\displaystyle\prod_{\alpha,\beta \in R_1}\left(1 - \alpha\overline{\beta}\right) \cdot \prod_{\alpha,\beta \in R_2}\left(1 - \alpha\overline{\beta}\right)}{\displaystyle\prod_{\alpha \in R_1, \beta \in R_2}|1 - \alpha\beta|^2}, \tag{7}$$

and, otherwise, $\mathscr{A}_q(\boldsymbol{D}_1, \boldsymbol{D}_2) = 0$. The detail of the derivation is in Appendix G.

Through this metric, we can observe a kind of "phase transition" of linear dynamical systems on the unit circle, and the metric has sensitive behavior when eigen values on it. We note that in the case of $P_i = Q_i = \emptyset$, the formula (7) is essentially equivalent to the distance (5) (see Theorem 4 in [6]).

## 4.2 Relation to the Binet-Cauchy metric on dynamical systems

Here, we discuss the relation between our metric and the Binet-Cauchy kernels on dynamical systems defined by Vishwanathan et al. [25, Section 5]. Let us consider two linear dynamical systems as in Subsection 4.1. In [25, Section 5], they give two kernels to measure the distance between two systems (for simplicity, here we disregard the expectations over variables); the trace kernels $k_{\mathrm{tr}}$ and the determinant kernels $k_{\mathrm{det}}$, which are respectively defined by

$$k_{\mathrm{tr}}((\boldsymbol{x}_{1,0}, \boldsymbol{f}_1, \boldsymbol{h}_1), (\boldsymbol{x}_{2,0}, \boldsymbol{f}_2, \boldsymbol{h}_2)) = \sum_{t=1}^{\infty} e^{-\lambda t}\boldsymbol{y}_{1,t}^{\top}\boldsymbol{y}_{2,t},$$

$$k_{\mathrm{det}}((\boldsymbol{x}_{1,0}, \boldsymbol{f}_1, \boldsymbol{h}_1), (\boldsymbol{x}_{2,0}, \boldsymbol{f}_2, \boldsymbol{h}_2)) = \det\left(\sum_{t=1}^{\infty} e^{-\lambda t}\boldsymbol{y}_{1,t}\boldsymbol{y}_{2,t}^{\top}\right),$$

where $\lambda > 0$ is a positive number satisfying $e^{-\lambda}||\boldsymbol{f}_1||||\boldsymbol{f}_2|| < 1$ to make the limits convergent. And $\boldsymbol{x}_{1,0}$ and $\boldsymbol{x}_{2,0}$ are initial state vectors, which affect the kernel values through the evolutions of the observation sequences. Vishwanathan et al. discussed a way of removing the effect of initial values by taking expectations over those by assuming some distributions.

These kernels can be described in terms of our notation as follows (see also Remark 3.3). That is, let us regard $\mathcal{H}_k = \mathbb{C}^q$. For $i = 1, 2$, we define $D_i := (e^{-\lambda} \boldsymbol{f}_i, \boldsymbol{h}_i, \boldsymbol{x}_{i,0}) \in \mathscr{T}(\mathbb{C}, \mathbb{C}^r)$, and $D_i^* := (e^{-\lambda} \boldsymbol{f}_i^*, \boldsymbol{x}_{i,0}^*, \boldsymbol{h}_i^*) \in \mathscr{T}(\mathbb{C}^r, \mathbb{C})$. Then these are described as

$$k_{\mathrm{tr}}\left((\boldsymbol{x}_{1,0}, \boldsymbol{f}_1, \boldsymbol{h}_1), (\boldsymbol{x}_{2,0}, \boldsymbol{f}_2, \boldsymbol{h}_2)\right) = \lim_{T \to \infty} \mathfrak{K}_1^T\left(D_1, D_2\right),$$

$$k_{\mathrm{det}}\left((x_{1,0}, \boldsymbol{f}_1, \boldsymbol{h}_1), (x_{2,0}, \boldsymbol{f}_2, \boldsymbol{h}_2)\right) = \lim_{T \to \infty} \mathfrak{K}_r^T\left(D_1^*, D_2^*\right).$$

Note that, introducing the exponential discounting $e^{-\lambda}$ is a way to construct a mathematically valid kernel to compare dynamical systems. However, in a certain situation, this method does not work effectively. In fact, if we consider three dynamical systems on $\mathbb{R}$: fix a small positive number $\epsilon > 0$ and let $f_1(x) = (1 + \epsilon)x$, $f_2(x) = x$, and $f_3(x) = (1 - \epsilon)x$ be linear dynamical systems. We choose $1 \in \mathbb{R}$ as the initial value. Here, it would be natural to regard these dynamical systems are "different" each other even with almost zero $\epsilon$. However, if we compute the kernel defined via the exponential discounting, these dynamical systems are judged to be similar or almost the same. Instead of introducing such an exponential discounting, our idea to construct a mathematically valid kernel is considering the limit of the ratio of kernels defined via finite series of the orbits of dynamical systems. As a consequence, we do not need to introduce the exponential discounting. It enables ones to deal with a wide range of dynamical systems, and capture the difference of the systems effectively. In fact, in the above example, our kernel judges these dynamical systems are completely different, i.e., the value of $A_1$ for each pair among them takes zero.

# 5 Empirical Evaluations

We empirically illustrate how our metric works with synthetic data of the rotation dynamics on the unit disk in a complex plane in Subsection 5.1, and then evaluate the discriminate performance of our metric with real-world time-series data in Subsection 5.2.

## 5.1 Illustrative example: Rotation on the unit disk

We use the rotation dynamics on the unit disk in the complex plane since we can compute the analytic solution of our metric for this dynamics. Here, we regard $\mathcal{X} = \mathbb{D} := \{z \in \mathbb{C} \mid |z| < 1\}$ and let $k(z, w) := (1 - z\overline{w})^{-1}$ be the Szegö kernel for $z, w \in \mathbb{D}$. The corresponding RKHS $\mathcal{H}_k$ is the space of holomorphic functions $f$ on $\mathbb{D}$ with the Taylor expansion $f(z) = \sum_{n \geq 0} a_n(f) z^n$ such that $\sum_{n \geq 0} |a_n(f)|^2 < \infty$. For $f, g \in \mathcal{H}_k$, the inner product is defined by $\langle f, g \rangle := \sum_{n \geq 0} a_n(f)\overline{a_n(g)}$. Let $\mathcal{H}_{\mathrm{in}} = \mathbb{C}$ and $\mathcal{H}_{\mathrm{ob}} = \mathcal{H}_k$.

For $\alpha \in \mathbb{C}$ with $|\alpha| \leq 1$, let $R_\alpha : \mathbb{D} \to \mathbb{D}; z \mapsto \alpha z$. We denote by $K_\alpha$ the Koopman operator for RKHS defined by $R_\alpha$. We note that since $K_\alpha$ is the adjoint of the composition operator defined by $R_\alpha$, by Littlewood subordination theorem, $K_\alpha$ is bounded. Now, we define $\delta_z : \mathcal{H}_k \to \mathbb{C}; f \mapsto f(z)$ and $\delta_{z,w} : \mathcal{H}_k \to \mathbb{C}^2; f \mapsto (f(z), f(w))$. Then we define $D_{\alpha,z}^1 := (R_\alpha, \phi, \delta_z^*) \in \mathscr{T}(\mathbb{C}, \mathcal{H}_k)$ and $D_{\alpha,z}^2 := (R_\alpha, \phi, \delta_{z,\alpha z}^*) \in \mathscr{T}(\mathbb{C}^2, \mathcal{H}_k)$.

By direct computation, we have the following formula (see Appendix H and Appendix I for the derivation): For $\mathscr{A}_1$, we have

$$
\mathscr{A}_1\left(D_{\alpha,z}^1, D_{\beta,w}^1\right) = \begin{cases}
\frac{(1-|z|^2)(1-|w|^2)}{|1-(z\overline{w})^q|^2} & |\alpha| = |\beta| = 1 \text{ and } \alpha\overline{\beta} = e^{2\pi i p/q} \text{ with } (p,q) = 1, \\
(1-|z|^2)(1-|w|^2) & |\alpha| = |\beta| = 1 \text{ and } \alpha\overline{\beta} = e^{2\pi i \gamma} \text{ with } \gamma \notin \mathbb{Q}, \\
1 - |z|^2 & |\alpha| = 1, |\beta| < 1, \\
1 - |w|^2 & |\alpha| < 1, |\beta| = 1, \\
1 & |\alpha|, |\beta| < 1.
\end{cases}
\tag{8}
$$

For $A_2$ we have

$$
\mathscr{A}_2\left(D_{\alpha,z}^2, D_{\beta,w}^2\right) = \begin{cases}
O(|zw|^{2\mu(\alpha,\beta)}) & |\alpha| = |\beta| = 1 \\
0 & |\alpha| = 1, |\beta| < 1, \\
0 & |\alpha| < 1, |\beta| = 1, \\
\frac{(1-|\alpha|^2)(1-|\beta|^2)}{|1-\alpha\overline{\beta}|^2} \cdot \frac{|1+\alpha\overline{\beta}|^2}{(1+|\alpha|^2)(1+|\beta|^2)} + O(|z\overline{w}|^2) & |\alpha|, |\beta| < 1.
\end{cases}
\tag{9}
$$

where, $\mu(\alpha, \beta)$ is a positive scalar value described in Appendix I. From the above, we see that $A_1$ depends on the initial values of $z$ and $w$, but $A_2$ could independently discriminate the dynamics.

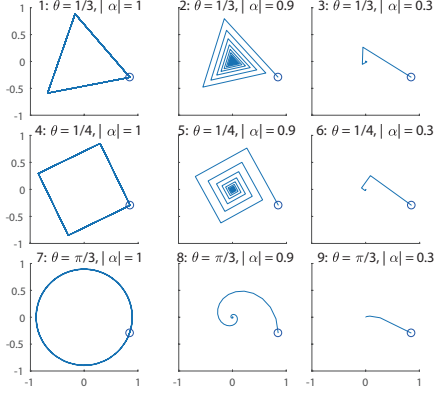

Figure 1: Orbits of rotation dynamics by multiplying $\alpha = |\alpha|e^{2\pi i\theta}$ on the unit disk with the same initial values.

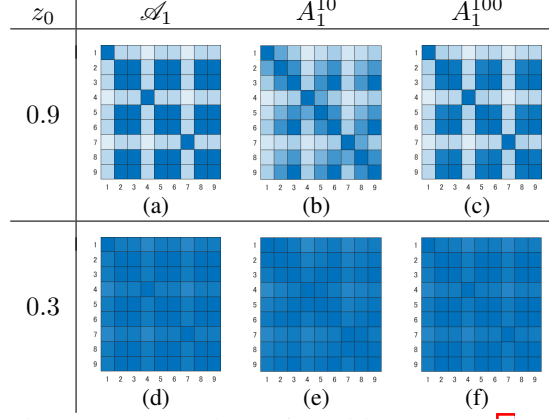

Figure 2: Comparison of empirical values (4) and theoretical values (8) of the kernels $A_1^T$ and $\mathscr{A}_1$ of rotation dynamics with initial values $z_0$

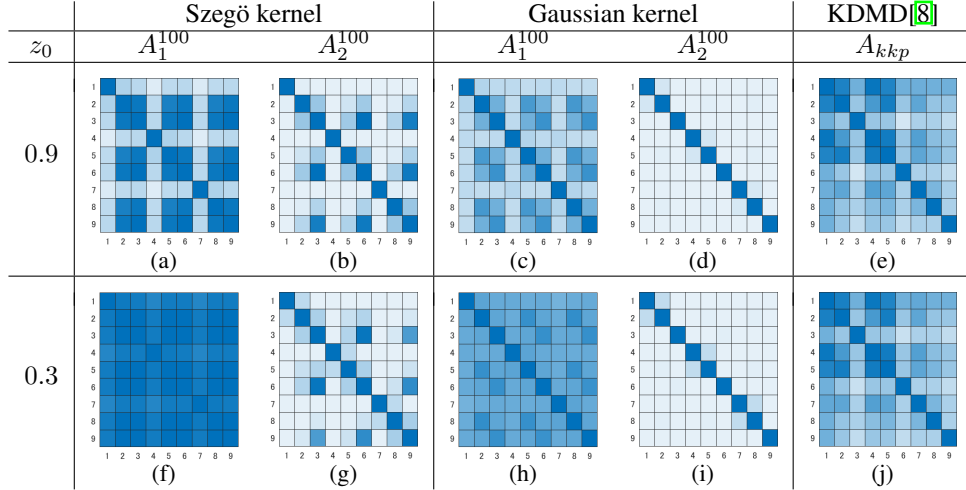

Figure 3: Discrimination results of various metrics for rotation dynamics with initial values $z_0$. Vertical and horizontal axes correspond to the dynamics in Figure 1.

Next, we show empirical results with Eq. (4) from finite data for this example.[1] For $\mathscr{A}_1$, we consider $x_{\alpha,t}^1 = \alpha^t z_0$, where $\alpha = |\alpha|e^{2\pi i\theta}$. And for $\mathscr{A}_2$, we consider $x_{\alpha,t}^1 = \alpha^t z_0$ and $x_{\alpha,t}^2 = \alpha^{t+1} z_0 = \alpha^t z_1$. The graphs in Figure 1 show the dynamics on the unit disk with $\theta = \{1/3, 1/4, \pi/3\}$ and $|\alpha| = \{1, 0.9, 0.3\}$. For simplicity, all of the initial values were set so that $|z_0| = 0.9$.

Figure 3 shows the confusion matrices for the above dynamics to see the discriminative performances of the proposed metric using the Szegö kernel (Figure 3a, 3b, 3f, and 3g), using radial basis function (Gaussian) kernel (Figure 3c, 3d, 3h, and 3i), and the comparable previous metric (Figure 3e and 3j) [8]. For the Gaussian kernel, the kernel width was set as the median of the distances from data. The last metric called Koopman spectral kernels [8] generalized the kernel defined by Vishwanathan et al. [25] to the nonlinear dynamical systems and outperformed the method. Among the above kernels, we used Koopman kernel of principal angle ($A_{kkp}$) between the subspaces of the estimated Koopman mode, showing the best discriminative performance [8].

The discriminative performance in $A_1$ when $T = 100$ shown in Figure 2c converged to the analytic solution when considering $T \to \infty$ in Figure 2a compared with that when $T = 10$ in Figure 2b. As guessed from the theoretical results, although $A_1$ did not discriminate the difference between the dynamics converging to the origin while rotating and that converging linearly, $A_2$ in Figure 3b did. $A_2$ using the Gaussian kernel ($A_{g2}$) in Figure 3d achieved almost perfect discrimination, whereas $A_1$ using Gaussian kernel ($A_{g1}$) in Figure 3c and $A_{kkp}$ in Figure 3e did not. Also, we examined the

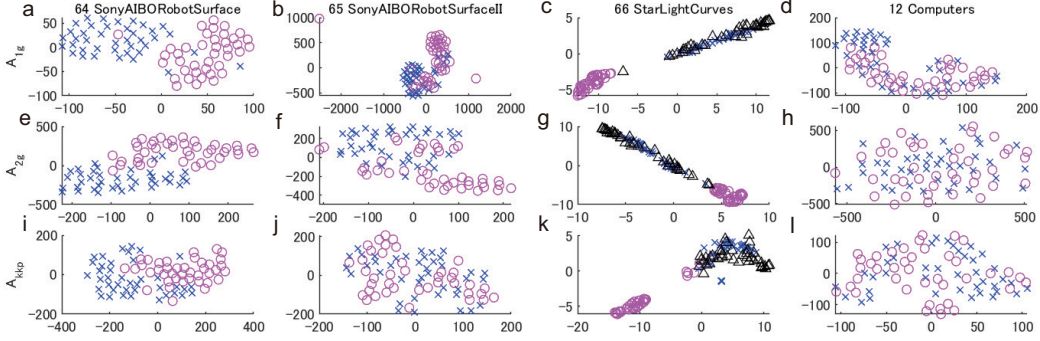

Figure 4: Embeddings of four time series data using t-SNE for $A_{g1}$ (a-d), $A_{g2}$ (e-h), and $A_{kkp}$ (i-l). (a,e,i) Sony AIBO robot surface I and (b,f,j) II datasets. (c,g,k) Star light curve dataset. (d,h,l) Computers dataset. The markers x, o, and triangle represent the class 1, 2, and 3 in the datasets.

case of small initial values in Figure 3f–3j so that $|z_0| = 0.3$ for all the dynamics. $A_2$ (Figure 3g, 3i) discriminated the two dynamics, whereas the remaining metrics did not (Figure 3f, 3h, and 3j).

## 5.2 Real-world time-series data

In this section, we evaluated our algorithm for discrimination using dynamical properties in time-series datasets from various real-world domains. We used the UCR time series classification archive as open-source real-world data [5]. It should be noted that our algorithm in this paper primarily target the deterministic dynamics; therefore, we selected the examples apparently with smaller noises and derived from some dynamics (For random dynamical systems, see e.g., [13, 26, 23]). From the above viewpoints, we selected two Sony AIBO robot surface (sensor data), star light curve (sensor data), computers (device data) datasets. We used $\mathbf{A}_m$ by Proposition 3.6 because we confirmed that the data satisfying the semi-stable condition in Definition 3.5 using the approximation of $K_f$ defined in [9].

We compared the discriminative performances by embedding of the distance matrices computed by the proposed metric and the conventional Koopman spectral kernel used above. For clear visualization, we randomly selected 20 sequences for each label from validation data, because our algorithms do not learn any hyper-parameters using training data. All of these data are one-dimensional time-series but for comparison, we used time-delay coordinates to create two-dimensional augmented time-series matrices. Note that it would be difficult to apply the basic estimation methods of Koopman modes assuming high-dimensional data, such as DMD and its variants. In addition, we evaluated the classification error using $k$-nearest neighbor classifier ($k = 3$) for simplicity. We used 40 sequences for each label and computed averaged 10-fold cross-validation error (over 10 random trials).

Figure 4 shows examples of the embedding of the $A_{g1}$, $A_{g2}$, and $A_{kkp}$ using t-SNE [24] for four time-series data. In the Sony AIBO robot surface datasets, D in Figure 4a,b,e,f (classification error: 0.025, 0.038, 0.213, and 0.150) had better discriminative performance than $A_{kkp}$ in Figure 4i,j (0.100 and 0.275). This tendency was also observed in the star light curve dataset in Figure 4c,g,k (0.150, 0.150, and 0.217), where one class (circle) was perfectly discriminated using $A_{g1}$ and $A_{g2}$ but the distinction in the remaining two class was less obvious. In computers dataset, $A_{g2}$, and $A_{kkp}$ in Figure 4h,l (0.450 and 0.450) show slightly better discrimination than $A_{kkp}$ in Figure 4d (0.500).

## 6 Conclusions

In this paper, we developed a general metric for comparing nonlinear dynamical systems that is defined with Koopman operator in RKHSs. We described that our metric includes Martin's metric and Binet-Cauchy kernels for dynamical systems as its special cases. We also described the estimation of our metric from finite data. Finally, we empirically showed the effectiveness of our metric using an example of rotation dynamics in a unit disk in a complex plane and real-world time-series data.

Several perspectives to be further investigated related to this work would exist. For example, it would be interesting to see discriminate properties of the metric in more details with specific algorithms. Also, it would be important to develop models for prediction or dimensionality reduction for nonlinear time-series data based on mathematical schemes developed in this paper.

## Footnotes

[1]The Matlab code is available at https://github.com/keisuke198619/metricNLDS

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
