[Supplementary Material]

## Supplementary Document for
## "Metric on Nonlinear Dynamical Systems with Koopman Operators"

In this supplementary material, we briefly explain the notion of the exterior product of Hilbert spaces in Section A, and then give the proofs of Proposition 3.1, Proposition 3.2, Proposition 3.6, Lemma 3.7, and Proposition 4.1 in Section D, Section B, Section C, Section E and Section F, respectively. And then, we describe the details of the derivations of Eq. (7), and the analytic solutions of $A_1$ in Eq. (8) and $A_2$ in Eq. (9) in Section G, Section H, and Section I, respectively.

## A  Exterior product of Hilbert spaces

Let $H$ be a Hilbert space with inner product $\langle \cdot, \cdot \rangle$. Let $H^{\otimes m}$ be a $m$-tensor product as an abstract complex linear space. Then for $x_1 \otimes \ldots x_m, y_1 \otimes \ldots y_m \in H^{\otimes m}$,

$$\langle x_1 \otimes \ldots x_m, y_1 \otimes \ldots y_m \rangle_\otimes := \prod_{i=1}^{m} \langle x_i, y_i \rangle$$

induces an inner product on $H^{\otimes m}$. We denote by $\widehat{\otimes}^m H$ the completion via the norm induced by the inner product $\langle \cdot, \cdot \rangle_\otimes$.

We define a linear operator $\mathscr{E} : \widehat{\otimes}^m H \to \widehat{\otimes}^m H$ by

$$\mathscr{E}(x_1 \otimes x_m) := \sum_{\sigma \in \mathfrak{S}_m} \mathrm{sgn}(\sigma) x_{\sigma(1)} \otimes \cdots \otimes x_{\sigma(m)},$$

where $\mathfrak{S}_m$ is the $m$-th symmetric group, and $\mathrm{sgn} : \mathfrak{S}_m \to \{\pm 1\}$ is the sign homomorphism. We define the $m$-*th exterior product* of $H$ by

$$\bigwedge^m H := \mathscr{E}\left(\widehat{\otimes}^m H\right).$$

For $x_1, \ldots, x_m \in H$, we also define

$$x_1 \wedge \ldots x_m := \mathscr{E}(x_1 \otimes \cdots \otimes x_m)$$

The inner product on $\bigwedge^m H$ is described as

$$\langle x_1 \wedge \cdots \wedge x_m, y_1 \wedge \cdots \wedge y_m \rangle_{\bigwedge^m H} = \det(\langle x_i, y_j \rangle)_{i,j=1,\ldots,m}.$$

We note that there exists an isomorphism

$$\bigoplus_{r+s=m} \bigwedge^r H \widehat{\otimes} \bigwedge^s H' \cong \bigwedge^m (H \oplus H'); \quad \sum_{r+s=m} x_r \otimes y_s \mapsto \sum_{r+s=m} x_r \wedge y_s.$$

Let $L : H \to H'$ be a linear operator. Then $L$ induces a linear operator $\widehat{\otimes}^m L : \widehat{\otimes}^m H \to \widehat{\otimes}^m H'$ defined by $\widehat{\otimes}^m L(x_1 \otimes \ldots x_m) := Lx_1 \otimes \cdots \otimes Lx_m$. The operator $\widehat{\otimes}^m L$ induces an operator on $\bigwedge^m H$, namely, $\widehat{\otimes}^m L (\bigwedge^m H) \subset \bigwedge^m H'$, and we define

$$\bigwedge^m L := \widehat{\otimes}^m L \Big|_{\bigwedge^m H}.$$

## B  Proof of Proposition 3.1

*Proof.* Let $S(D_i) := \left(L_{h_i} \mathscr{I}_i, \ldots, L_{h_i} K_{\boldsymbol{f}_i}^{T-1} \mathscr{I} i\right)$, which is a linear operator from $\mathcal{H}_{\mathrm{in}}$ to $\mathcal{H}_{\mathrm{ob}}^T$. Since $\wedge^m S(D_2)^* S(D_1) = (\wedge^m S(D_2))^* (\wedge^m S(D_1))$, $\mathfrak{K}_m(D_1, D_2)$ is just a inner product of the Hilbert-Schmidt operators $\wedge^m S(D_1)$ nad $\wedge^m S(D_1)$, thus, we see that $\mathfrak{K}_m^T$ is a positive definite kernel. $\square$

# C   Proof of Proposition 3.2

*Proof.* Since $\mathscr{A}_m^{\mathcal{B}}$ is a positive definite kernel on $\mathscr{T}(\mathcal{H}_{\mathrm{in}}, \mathcal{H}_{\mathrm{ob}})$, there exists RKHS $\mathcal{H}_{\mathscr{A}_m^{\mathcal{B}}}$ with feature map $\tilde{\phi} : \mathscr{T}(\mathcal{H}_{\mathrm{in}}, \mathcal{H}_{\mathrm{ob}}) \to \mathcal{H}_{\mathscr{A}_m^{\mathcal{B}}}$. Therefore, the statement of the theorem follows that

$$\sqrt{1 - \mathscr{A}_m^{\mathcal{B}}(D_1, D_2)} = 2^{-1/2} \left\| \tilde{\phi}(D_1) - \tilde{\phi}(D_2) \right\|_{\mathcal{H}_{\mathscr{A}_m^{\mathcal{B}}}}$$

$\square$

# D   Proof of Proposition 3.6

In this section, we denote $v_r := v_{1,r}$, and $w_r := v_{2,r}$. Also, let $\varphi_{n_1,\dots,n_m} := a_{1,n_1} L_{h_1} \varphi_{1,n_1} \wedge \cdots \wedge a_{1,n_m} L_{h_1} \varphi_{1,n_m}$ and $\psi_{n_1,\dots,n_m} := a_{2,n_1} L_{h_2} \psi_{2,n_1} \wedge \cdots \wedge a_{2,n_m} L_{h_1} \psi_{2,n_m}$. Then we have

$$\left\langle L_{h_1} K_{\boldsymbol{f}_1}^{t_1} v_{s_1} \wedge \cdots \wedge L_{h_1} K_{\boldsymbol{f}_1}^{t_m} v_{s_m}, \; L_{h_2} K_{\boldsymbol{f}_2}^{t_1} w_{s_1} \wedge \cdots \wedge L_{h_2} K_{\boldsymbol{f}_2}^{t_m} w_{s_m} \right\rangle$$

$$= \sum_{\substack{p_1,\dots,p_m=1 \\ q_1,\dots,q_m=1}}^{\infty} \lambda_{1,p_1}^{t_1} \overline{\lambda_{2,q_1}}^{t_1} \cdots \lambda_{1,p_m}^{t_m} \overline{\lambda_{2,q_m}}^{t_m} \left\langle \varphi_{p_1,\dots,p_m}, \; \psi_{q_1,\dots,q_m} \right\rangle .$$

Thus we have the following formulas:

$$\mathfrak{K}_m^T(D_1, D_2)$$
$$= \sum_{\substack{p_1,\dots,p_m=1 \\ q_1,\dots,q_m=1}}^{\infty} \frac{1 - \lambda_{1,p_1}^T \overline{\lambda_{2,q_1}}^T}{1 - \lambda_{1,p_1} \overline{\lambda_{2,q_1}}} \cdots \frac{1 - \lambda_{1,p_m}^T \overline{\lambda_{2,q_m}}^T}{1 - \lambda_{1,p_m} \overline{\lambda_{2,q_m}}} \left\langle \varphi_{p_1,\dots,p_m}, \; \psi_{q_1,\dots,q_m} \right\rangle ,$$

$$\mathfrak{K}_m^T(D_1, D_1)$$
$$= \sum_{\substack{p_1,\dots,p_m=1 \\ q_1,\dots,q_m=1}}^{\infty} \frac{1 - \lambda_{1,p_1}^T \overline{\lambda_{1,q_1}}^T}{1 - \lambda_{1,p_1} \overline{\lambda_{1,q_1}}} \cdots \frac{1 - \lambda_{1,p_m}^T \overline{\lambda_{1,q_m}}^T}{1 - \lambda_{1,p_m} \overline{\lambda_{1,q_m}}} \left\langle \varphi_{p_1,\dots,p_m}, \; \varphi_{q_1,\dots,q_m} \right\rangle ,$$

$$\mathfrak{K}_m^T(D_2, D_2)$$
$$= \sum_{\substack{p_1,\dots,p_m=1 \\ q_1,\dots,q_m=1}}^{\infty} \frac{1 - \lambda_{2,p_1}^T \overline{\lambda_{2,q_1}}^T}{1 - \lambda_{2,p_1} \overline{\lambda_{2,q_1}}} \cdots \frac{1 - \lambda_{2,p_m}^T \overline{\lambda_{2,q_m}}^T}{1 - \lambda_{2,p_m} \overline{\lambda_{2,q_m}}} \left\langle \psi_{p_1,\dots,p_m}, \; \psi_{q_1,\dots,q_m} \right\rangle ,$$

Here, for any complex number $z$, if $z = 1$, we regard $(1 - z^T)/(1 - z)$ as $T$.

We may assume both $\mathfrak{K}_m^T(D_1, D_1)$ and $\mathfrak{K}_m^T(D_2, D_2)$ are grater than some positive constant not depending on $T$.

At first, we treat the case $D_1$ and $D_2$ are semi-stable. In this case, we see that $\mathfrak{K}_m^T(D_i, D_j) = c_{i,j} T^{n_{ij}} + o(T^{n_{ij}})$ for some some constant $c_{i,j}$ and non-negative integer $n_{ij} \geq 0$. Moreover, we have $2n_{12} \leq n_{11}, n_{22}$. By combining this with that $\sum_{n_1,\dots,n_m} \|\varphi_{n_1,\dots,n_m}\|$ and $\sum_{n_1,\dots,n_m} \|\psi_{n_1,\dots,n_m}\|$ converge, we see that $\mathbf{A}_m$ converges and the limit is equal to $\mathscr{A}_m^{\mathcal{B}}$ for any Banach limit $\mathcal{B}$.

Next, put

$$K_{1,N}^T := \sum_{\substack{p_1,\ldots,p_m=1 \\ q_1,\ldots,q_m=1}}^N \frac{1 - \lambda_{1,p_1}^T \overline{\lambda_{2,q_1}}^T}{1 - \lambda_{1,p_1}\overline{\lambda_{2,q_1}}} \cdots \frac{1 - \lambda_{1,p_m}^T \overline{\lambda_{2,q_m}}^T}{1 - \lambda_{1,p_m}\overline{\lambda_{2,q_m}}} \langle \varphi_{p_1,\ldots,p_m}, \psi_{q_1,\ldots,q_m} \rangle,$$

$$K_{2,N}^T := \sum_{\substack{p_1,\ldots,p_m=1 \\ q_1,\ldots,q_m=1}}^N \frac{1 - \lambda_{1,p_1}^T \overline{\lambda_{1,q_1}}^T}{1 - \lambda_{1,p_1}\overline{\lambda_{1,q_1}}} \cdots \frac{1 - \lambda_{1,p_m}^T \overline{\lambda_{1,q_m}}^T}{1 - \lambda_{1,p_m}\overline{\lambda_{1,q_m}}} \langle \varphi_{p_1,\ldots,p_m}, \varphi_{q_1,\ldots,q_m} \rangle,$$

$$K_{3,N}^T := \sum_{\substack{p_1,\ldots,p_m=1 \\ q_1,\ldots,q_m=1}}^N \frac{1 - \lambda_{2,p_1}^T \overline{\lambda_{2,q_1}}^T}{1 - \lambda_{2,p_1}\overline{\lambda_{2,q_1}}} \cdots \frac{1 - \lambda_{2,p_m}^T \overline{\lambda_{2,q_m}}^T}{1 - \lambda_{2,p_m}\overline{\lambda_{2,q_m}}} \langle \psi_{p_1,\ldots,p_m}, \psi_{q_1,\ldots,q_m} \rangle,$$

$$\mathcal{A}_N := \left( \frac{|K_{1,N}^T|^2}{K_{2,N}^T K_{3,N}^T} \right)_{T=1}^{\infty} \in \ell^{\infty},$$

Since $\mathcal{A}_N \to \mathbf{A}_m$ as $N \to \infty$, thus $C\mathcal{A}_N \to C\mathbf{A}_m$ and thus it suffices to show that $C\mathcal{A}_N$ converges for any sufficiently large $N$, but, the convergence of $C\mathcal{A}_N$ actually follows the Lemma D.1 below. $\mathcal{BC} = \mathcal{B}$.

**Lemma D.1.** *We denote by* $S := \left\{ z \in \mathbb{C} \mid |z| = 1 \right\}$ *the unit circle in* $\mathbb{C}$. *Let* $f : S^m \times \mathbb{C}^n \to \mathbb{C}$ *be a continuous function. Let* $\boldsymbol{\zeta} \in S^m$ *and* $\{\boldsymbol{x}_i\}_{i\geq 0} \subset \mathbb{C}^n$ *be a sequence convergent to zero. Then the limit*

$$\lim_{T \to \infty} \frac{1}{T} \sum_{t=1}^T f(\boldsymbol{\zeta}^t, \boldsymbol{x}_t)$$

*converges.*

*Proof.* By the Weierstrass' approximation theorem, we may assume $f$ is a $m + n$-variable monomial: $f(x_1, \ldots, x_{m+n}) = x_1^{r_1} \ldots x_{m+n}^{r_{m+n}}$. Thus $f(\boldsymbol{\zeta}^t, \boldsymbol{x}_t)$ is regarded as $\zeta^t$ or $\zeta^t y_t$ where $\zeta \in \mathbb{C}$ with $|\zeta| = 1$ and $\{\boldsymbol{y}_t\}_{t\geq 0}$ is a sequence convergent to zero. In the both cases, we see that the limit in the lemma exists. $\square$

# E  Proof of Lemma 3.7

We use the property of trace of a linear operator $A$ on $\mathbb{C}^N$:

$$\mathrm{tr}(A \wedge \cdots \wedge A) = \sum_{0 < s_1 < \cdots < s_m \leq N} \langle A\mathbf{e}_{s_1} \wedge \cdots \wedge A\mathbf{e}_{s_m}), \mathbf{e}_{s_1} \wedge \cdots \wedge \mathbf{e}_{s_m} \rangle,$$

$$= \sum_{0 < s_1 < \cdots < s_m \leq N} \langle \mathcal{E}(A\mathbf{e}_{s_1} \otimes \cdots \otimes A\mathbf{e}_{s_m}), \mathcal{E}(\mathbf{e}_{s_1} \otimes \cdots \otimes \mathbf{e}_{s_m}) \rangle.$$

Here, $\mathbf{e}_{s_k}$ be $N$-length vectors whose $s_k$-th component is 1 and the others are 0, and

$$\mathcal{E}(v_1 \otimes \ldots v_N) =: v_1 \wedge \cdots \wedge v_N = \frac{1}{N!} \sum_{\sigma \in \mathcal{S}_N} \mathrm{sgn}(\sigma) v_{\sigma(1)} \otimes \cdots \otimes v_{\sigma(N)}$$

where $\mathcal{S}_N$ is the symmeric group of degree $N$. Thus, we have

$$\mathfrak{K}_m^T((L_{h_i}, K_{\boldsymbol{f}_i}, X_i), (L_{h_j}, K_{\boldsymbol{f}_j}, X_j))$$

$$= \sum_{t_1,\ldots,t_m=0}^{T-1} \sum_{1 \leq s_1 < \cdots < s_m \leq N}$$

$$\left\langle L_{h_i} K_{\boldsymbol{f}_i}^{t_1} X_i^{(s_1)} \wedge \cdots \wedge L_{h_i} K_{\boldsymbol{f}_i}^{t_m} X_i^{(s_m)}, L_{h_j} K_{\boldsymbol{f}_j}^{t_1} X_j^{(s_1)} \wedge \cdots \wedge L_{h_j} K_{\boldsymbol{f}_j}^{t_m} X_j^{(s_m)} \right\rangle.$$

Here, we use the property of Hermite transpose of wedge product: $(A_1 \otimes \cdots \otimes A_m)^* = A_1^* \otimes \cdots \otimes A_m^*$ for operators $A_1, \ldots, A_m$.

# F  Proof of Proposition 4.1

Let $V_i$ be a matrix making $\boldsymbol{A}_i$ the Jordan normal form in the following form:

$$V_i^{-1}\boldsymbol{A}_i V_i = \begin{pmatrix} \widetilde{D}_i & \cdots & & 0 \\ & J_{i,n_{i,1}} & & \vdots \\ & & \ddots & \\ 0 & \cdots & & J_{i,n_{i,M_i'}} \end{pmatrix}.$$

Here, all the $n_{i,k} > 1$ and $\widetilde{D}_i := \mathrm{diag}(\alpha_{i,1},\ldots,\alpha_{i,M_i})$ and $J_{i,n_{i,k}} := \beta_{i,k}\boldsymbol{I}_{n_{i,k}} + N_{n_{i,k}-1}'$ where

$$N_r' := \begin{pmatrix} \boldsymbol{o} & \boldsymbol{I}_r \\ 0 & \boldsymbol{o}^\top \end{pmatrix}.$$

We assume

$$|\alpha_{i,1}| \geq \cdots \geq |\alpha_{i,l_i}| > 1 = |\alpha_{i,l_i+1}| = \cdots = |\alpha_{i,m_i}| > |\alpha_{i,m_i+1}| \geq \cdots \geq |\alpha_{i,M_i}|,$$
$$|\beta_{i,1}| \geq \cdots \geq |\beta_{i,l_i'}| > 1 = |\beta_{i,l_i'+1}| = \cdots = |\beta_{i,m_i'}| > |\beta_{i,m_i'+1}| \geq \cdots \geq |\beta_{i,M_i'}|$$

Let

$$N_i := \begin{pmatrix} \boldsymbol{0} & \cdots & & \boldsymbol{0} \\ \vdots & N_{n_{i,1}-1}' & & \vdots \\ & & \ddots & \\ \boldsymbol{0} & \cdots & & N_{n_{i,M_i'}-1}' \end{pmatrix},$$

be a nilpotent matrix and let $D_i := V_i^{-1}\boldsymbol{A}_i V_i - N_i$ be a diagonal matrix. Then direct computation shows that

$$\mathfrak{K}_q^T\left((L_{\boldsymbol{C}_i}, K_{\boldsymbol{A}_i}, \boldsymbol{I}_q), (L_{\boldsymbol{C}_j}, K_{\boldsymbol{A}_j}, \boldsymbol{I}_q)\right)$$
$$= \det V_i^{-1} \cdot \overline{\det V_j}^{-1} \det\left(\sum_{a,b=0}^{q}\sum_{r=0}^{T-1} {}_rC_a \cdot {}_rC_b \cdot D_j^{*r-b} N_j^{*b} W N_i^a D_i^{r-a}\right)$$

where $W = V_j^* \boldsymbol{C}_j^* \boldsymbol{C}_i V_i$. Put

$$B_{i,j,T} := \sum_{a,b=0}^{q}\sum_{r=0}^{T-1} {}_rC_a \cdot {}_rC_b \cdot D_j^{*r-b} N_j^{*b} W N_i^a D_i^{r-a}.$$

For $T > 0$, we define

$$D_{i,T}' := \mathrm{diag}\left(\alpha_{i,1}^{-T},\ldots,\alpha_{i,l_i}^{-T}, T_{l_i+1}^{-1/2},\ldots,T_{m_i}^{-1/2}, 1,\ldots,\frac{1}{M_i}\right.$$
$$,\beta_{i,1}^{-T}, T^{-1}\beta_{i,1}^{-T},\ldots,T^{-n_{i,1}+1}\beta_{i,1}^{-T},\ldots,\beta_{i,l_i'}^{-T},\ldots,T^{-n_{i,l_i'}+1}\beta_{i,l_i'}^{-T}$$
$$\left., T^{-1/2},\ldots,T^{1/2-n_{i,l_i+1}},\ldots,T^{1/2-n_{i,m_i'}}, 1,\ldots,1\right).$$

Then we see that $\lim_{T\to\infty} D_{j,T}'^* B_{i,j,T} D_{i,T}'$ exists. Therefore, since

$$A_q^T(\boldsymbol{D}_1, \boldsymbol{D}_2) = \frac{\left|\det\left(D_{2,T}'^* B_{1,2,T} D_{1,T}'\right)\right|^2}{\det\left(D_{1,T}'^* B_{1,1,T} D_{1,T}'\right)\det\left(D_{2,T}'^* B_{2,2,T} D_{2,T}'\right)},$$

the limit of $\boldsymbol{A}_q$ exists.

If the systems are stable and observable, it is the direct consequece of the definition of principal angles (see the formula (1) in [6]).

## G    Derivation of Eq. (7)

Let
$$P_i = \{\gamma_{i,1}, \ldots, \gamma_{i,l_i}\}, \ Q_i = \{\gamma_{i,l_i+1}, \ldots, \gamma_{i,m_i}\}, \text{ and } R_i = \{\gamma_{i,m_i+1}, \ldots, \gamma_{i,N_i}\}.$$

Define $D'_{i,T} := \mathrm{diag}\Big((\gamma_{i,1})^{-T}, \ldots, (\gamma_{i,l_i})^{-T}, \underset{l_i+1}{\sqrt{T}^{-1}}, \ldots, \underset{m_i}{\sqrt{T}^{-1}}, 1, \ldots, 1\Big)$. Then, for $i = 1, 2$, we have

$$\lim_{T\to\infty} \mathfrak{K}_q^T(\boldsymbol{D}_1, \boldsymbol{D}_2) \cdot |\det D'_{i,T}|^2 \cdot |\det V_i|^{-2}$$
$$= (-1)^{\#P_i} \det\left(\frac{1}{1-\alpha\overline{\beta}}\right)_{\alpha,\beta\in P_i} \det\left(\frac{1}{1-\alpha\overline{\beta}}\right)_{\alpha,\beta\in R_i}. \tag{10}$$

On the other hand,

$$\lim_{T\to\infty} \left|\mathfrak{K}_q^T(\boldsymbol{D}_1, \boldsymbol{D}_2) \cdot \det D'_{1,T} \cdot \overline{\det D'_{2,T}}\right| \cdot \left|\det V_1 \cdot \overline{\det V_2}\right|^{-1} \tag{11}$$

$$= \begin{cases} \left|\det\left(\dfrac{1}{1-\alpha\overline{\beta}}\right)_{\substack{\alpha\in P_1, \\ \beta\in P_2}}\right| \cdot \left|\det\left(\dfrac{1}{1-\alpha\overline{\beta}}\right)_{\substack{\alpha\in R_1, \\ \beta\in R_2}}\right| & \text{if } |P_1| = |P_2|, |R_1| = |R_2|, Q_1 = Q_2, \\ 0 & \text{otherwise.} \end{cases}$$

Here, we give a sketch of the proof of Eqs. (10) and (11). Since both are proved in a similar way, we only show Eq. (10) in the case of $i = 1$.

*Proof of (10) in the case of $i = 1$.*  Recall
$$\mathfrak{K}_N^T(\boldsymbol{D}_1, \boldsymbol{D}_2) = \det\left(\sum_{r=0}^{T-1} (V_1^{-1})^* D_1^{*r+1} W D_1^{r+1} V_1^{-1}\right)$$
$$= \left|\det D_1 V_1^{-1}\right|^2 \cdot \det\left(\sum_{r=1}^{T} (\overline{\gamma_{1,s}}\gamma_{1,t})^r\right)_{s,t=1,\ldots N}$$

and put
$$C_T := \left(\sum_{r=0}^{T-1} (\overline{\gamma_{i,s}}\gamma_{i,t})^r\right)_{s,t=1,\ldots N},$$

where $W \in \mathbb{R}^{q\times q}$ whose components are all 1. The matrix $D'^*_{i,T} C_T D'_{i,T}$ is described as the following matrix with nine sections:

$$D'^*_{1,T} C_T D'_{1,T} = \begin{pmatrix} (P_1 P_1) & (P_1 Q_1) & (P_1 R_1) \\ (Q_1 P_1) & (Q_1 Q_1) & (Q_1 R_1) \\ (R_1 P_1) & (R_1 Q_1) & (R_1 R_1) \end{pmatrix}$$

where each section has an explicit description, for example,

$$(P_1 P_1) = \left(\frac{\overline{\gamma_{1,s}}^{-T}\gamma_{1,t}^{-T} - 1}{1 - \overline{\gamma_{1,s}}\gamma_{1,t}}\right)_{s,t=1,\ldots,l_1},$$

$$(P_1 Q_1) = \left(\frac{\overline{\gamma_{1,s}}^{-T} - \gamma_{1,t}^{T}}{\sqrt{T}(1 - \overline{\gamma_{1,s}}\gamma_{1,t})}\right)_{\substack{s=1,\ldots,l_1 \\ t=1,\ldots,m_1}}.$$

Thus we see that
$$\lim_{T\to\infty} (P_1 P_1) = \left(\frac{-1}{1-\alpha\overline{\beta}}\right)_{\alpha,\beta\in P_1}$$
$$\lim_{T\to\infty} (Q_1 Q_1) = \boldsymbol{I}_{m_1}$$
$$\lim_{T\to\infty} (R_1 R_1) = \left(\frac{1}{1-\alpha\overline{\beta}}\right)_{\alpha,\beta\in R_1}$$
$$\lim_{T\to\infty} (P_1 Q_1) = \lim_{T\to\infty} (P_1 R_1) = \lim_{T\to\infty} (Q_1 R_1) = 0$$

Therefore, we have

$$\lim_{T\to\infty} \mathfrak{K}_N^T(\boldsymbol{D}_1, \boldsymbol{D}_2) \cdot |\det D'_{1,T}|^2 \cdot |\det D_1 V_1^{-1}|^2$$

$$= (-1)^{\#P_1} \det\left(\frac{1}{1-\alpha\overline{\beta}}\right)_{\alpha,\beta\in P_1} \det\left(\frac{1}{1-\alpha\overline{\beta}}\right)_{\alpha,\beta\in R_1}.$$

$\square$

Also, for distinct complex numbers $x_1, \ldots, x_m, y_1, \ldots, y_m$, the determinant of the Cauchy matrix $\det\left((x_i - y_j)^{-1}\right)_{i,j=1,\ldots,m}$ is equal to

$$\frac{\prod_{i<j}(x_i - x_j)(y_j - y_i)}{\prod_{i,j}(x_i - y_j)}.$$

Combining it with

$$\det\left(\frac{1}{1-\alpha\overline{\beta}}\right)_{\substack{\alpha\in P_i,\\ \beta\in P_j}} = \det\left((\alpha^{-1} - \overline{\beta})^{-1}\right)_{\substack{\alpha\in P_i,\\ \beta\in P_j}} \prod_{\alpha\in P_i} \alpha^{-1}$$

and the similar formula for $\det\left((1 - \alpha\overline{\beta})^{-1}\right)_{\substack{\alpha\in R_i,\\ \beta\in R_j}}$, if $|P_1| = |P_2|$, $|R_1| = |R_2|$ and $Q_1 = Q_2$, we have Eq. (7). Otherwise, $\mathscr{A}_q(\boldsymbol{D}_1, \boldsymbol{D}_2) = 0$.

# H   Analytic solution of $\mathscr{A}_1$ (Eq. (8)) using Szegö kernel

In this appendix, we show the derivation of

$$\lim_{T\to\infty} \frac{1}{T}\mathfrak{K}_1^T(D^1_{\alpha,z}, D^1_{\beta,w}) = \lim_{T\to\infty} \frac{1}{T}\sum_{t=0}^{\infty} k(x^1(t), x^2(t))$$

$$= \lim_{T\to\infty} \frac{1}{T}\sum_{t=0}^{\infty} \frac{1}{1-(\alpha\overline{\beta})^t z\overline{w}}$$

$$= \begin{cases} \frac{1}{1-(z\overline{w})^q} & |\alpha| = |\beta| = 1 \text{ and } \alpha\overline{\beta} = e^{2\pi i p/q}, \\ 1 & \text{otherwise,} \end{cases}$$

where $p, q$ is relatively prime integers and let $q = +\infty$ when $\alpha\overline{\beta}$ rotates an irrational angle.

Here it suffices to consider the case of $|\alpha| = |\beta| = 1$ and $\alpha\overline{\beta}$ rotates a rational and irrational angles. Now, we set $\gamma = \alpha\overline{\beta}$ and $T' = z\overline{w}$. First, we consider the rational angle case. Then we will show the derivation of

$$\lim_{T\to\infty} \frac{1}{T}\sum_{t=0}^{T} \frac{1}{1-\gamma^t T'} = \frac{1}{1-T'^q}. \tag{12}$$

First, we will show the following proposition:

**Proposition H.1.** *Assume $\gamma = e^{2\pi i p/q}$, where $p, q$ is relatively prime integers and $T'$ is a constant complex value. Then, we have*

$$\sum_{t=0}^{q-1} \frac{1}{1-\gamma^t T'} = \frac{q}{1-T'^q}. \tag{13}$$

*Proof.* First, we remember the following fact:

$$\frac{q}{1-T'^q} = \sum_{t=0}^{q-1} \frac{a_t}{1-\gamma^t T'}, \tag{14}$$

where $a_t$ is a scalar coefficient for $t = \{0, \ldots, q-1\}$, which is calculated below. Here, we use the property: $1 - T'^{q-1} = \prod_{t=0}^{q-1} 1 - \gamma^t T'$ when $\gamma = e^{2\pi i p/q}$ (i.e., $\gamma^q = 1$). Then, for deriving the following solution

$$\sum_{t=0}^{q-1} \frac{a_t}{1 - \gamma^t T'} = \sum_{t=0}^{q-1} \frac{1}{1 - \gamma^t T'}, \tag{15}$$

we consider the limit on $T' \to \gamma^{-s}$ for $s = \{0, \ldots, q-1\}$ as follows:

$$\lim_{T' \to \gamma^{-s}} (1 - \gamma^s T') \sum_{t=0}^{q-1} \frac{1}{1 - \gamma^t T'} = \lim_{T' \to \gamma^{-s}} (1 - \gamma^s T') \sum_{t \neq s}^{q-1} \frac{a_t}{1 - \gamma^t T'} + a_s = a_s.$$

Thus, it is enough to calculate $a_s$, which is calculated as follows:

$$a_s = \lim_{T' \to r^{-s}} (1 - r^s T') \frac{q}{1 - T'^q} = (\gamma^s q) \lim_{T' \to r^{-s}} \left( \frac{T'^q - (\gamma^{-s})^q}{T' - \gamma^{-s}} \right)^{-1} = \gamma^s q \left( q(\gamma^{-s})^{q-1} \right)^{-1} = 1,$$

which gives Eq. (15). Therefore we obtain Eq. (13). $\qquad\square$

Next, we consider the limit on the time $T$. Consider $T = q b_T + c$, where $q, b_T, c$ are non-negative integers, $c < q$ and $b_T$ is a variable that changes with $T$, then we have

$$\lim_{T \to \infty} \frac{1}{T} \sum_{t=0}^{T} \frac{1}{1 - \gamma^t T'} = \lim_{T \to \infty} \frac{1}{T} \left( \sum_{t=0}^{c} \frac{1}{1 - \gamma^t T'} + \sum_{t=0}^{q b_T - 1} \frac{1}{1 - \gamma^t T'} \right)$$
$$= \lim_{T \to \infty} \frac{b_T}{T} \sum_{t=0}^{q-1} \frac{1}{1 - \gamma^t T'} = \frac{1}{1 - T'^q}, \tag{16}$$

which implies Eq. (12).

In case of $|\alpha| = |\beta| = 1$ and $\gamma = \alpha \overline{\beta}$ rotating an irrational angle, we set $\gamma = e^{2\pi i \xi}$, where $\xi$ is a irrational number. In complex analysis, we introduce the following fact:

$$\lim_{T \to \infty} \frac{1}{T} \sum_{t=0}^{\infty} \mathcal{F}(e^{2\pi i \xi t}) = \int_0^{2\pi} \mathcal{F}(e^{2\pi i \theta}) d\theta, \tag{17}$$

for any continuous function $\mathcal{F}$. By combining (17) and the residue theorem in complex analysis, we obtain

$$\lim_{T \to \infty} \frac{1}{T} \sum_{t=0}^{\infty} \frac{1}{1 - \gamma^t T'} = \int_0^{2\pi} \frac{1}{1 - e^{2\pi i \theta} T'} d\theta = \frac{1}{2\pi i} \int_{|x|=1} \frac{1}{(1 - xT')x} dx = 1.$$

# I  Analytic solution of $\mathscr{A}_2$ (Eq. (9)) using Szegö kernel

In this appendix, we show the derivation of

$$\mathscr{A}_2 \left( D_{z,\alpha}^2, D_{w,\beta}^2 \right) = \begin{cases} O(|zw|^{2\mu(\alpha,\beta)}) & |\alpha| = |\beta| = 1, \\ 0 & |\alpha| = 1, |\beta| < 1, \\ 0 & |\alpha| < 1, |\beta| = 1, \\ \frac{(1-|\alpha|^2)(1-|\beta|^2)}{|1-\alpha\overline{\beta}|^2} \cdot \frac{|1+\alpha\overline{\beta}|^2}{(1+|\alpha|^2)(1+|\beta|^2)} + O(|z\overline{w}|^2) & |\alpha|, |\beta| < 1, \end{cases}$$

where, for $\alpha = e^{2\pi i a}$ and $\beta = e^{2\pi i b}$, the integer $\mu(\alpha, \beta)$ is defined by

$$\mu(\alpha, \beta) = \begin{cases} q & a \notin \mathbb{Q} \text{ or } b \notin \mathbb{Q} \text{ with } a - b = p/q \text{ with } (p, q) = 1, \\ +\infty & a \notin \mathbb{Q} \text{ or } b \notin \mathbb{Q} \text{ with } a - b \notin \mathbb{Q}, \\ \min \{p + q \mid p, q \geq 0, ap - bq \in \mathbb{Z}\} & a, b \in \mathbb{Q}. \end{cases}$$

Let $\varphi_n = z^n \in \mathcal{H}_k$ be an element of RKHS. We note that $\{\varphi_n\}_{n=0}^{\infty}$ is an orthonomal basis. Moreover, since $K_\alpha$ is the adjoint of the composition operator of $R_\alpha$, we have:

$$K_\alpha \varphi_n = \overline{\alpha}^n \varphi_n.$$

As in the proof of Proposition 3.6 in Appedinx D, we have

$$
\begin{aligned}
& \mathfrak{K}_2^T \left( D_{z,\alpha}^2, D_{w,\beta}^2 \right) \\
&= \sum_{p_1,p_2,q_1,q_2=0}^{\infty} \frac{1 - \overline{\alpha}^{p_1 T} \beta^{q_1 T}}{1 - \overline{\alpha}^{p_1} \beta^{q_1}} \cdot \frac{1 - \overline{\alpha}^{p_2 T} \beta^{q_2 T}}{1 - \overline{\alpha}^{p_2} \beta^{q_2}} \overline{\alpha^{p_2} z^{p_1 + p_2}} \beta^{q_2} w^{q_1 + q_2} \langle \varphi_{p_1} \wedge \varphi_{p_2}, \varphi_{q_1} \wedge \varphi_{q_2} \rangle \\
&= \sum_{\substack{p,q=0 \\ p \neq q}}^{\infty} \frac{1 - \overline{\alpha}^{pT} \beta^{pT}}{1 - \overline{\alpha}^p \beta^p} \cdot \frac{1 - \overline{\alpha}^{qT} \beta^{qT}}{1 - \overline{\alpha}^q \beta^q} (\overline{\alpha}\beta)^q (\overline{z}w)^{p+q} \\
& \quad - \sum_{\substack{p,q=0 \\ p \neq q}}^{\infty} \frac{1 - \overline{\alpha}^{pT} \beta^{qT}}{1 - \overline{\alpha}^p \beta^q} \cdot \frac{1 - \overline{\alpha}^{qT} \beta^{pT}}{1 - \overline{\alpha}^q \beta^p} \overline{\alpha^q} \beta^p (\overline{z}w)^{p+q}.
\end{aligned}
$$

In particular, we see that

$$
\mathfrak{K}_2^T \left( D_{z,\alpha}^2, D_{w,\beta}^2 \right) = \begin{cases} O(T^2) & \text{if } |\alpha| = |\beta| = 1, \\ O(T) & \text{if } |\alpha\beta| < 1. \end{cases}
$$

Thus in the case of $|\alpha| = 1, |\beta| < 1$ or $|\beta| = 1, |\alpha| < 1$, we have

$$
\mathscr{A}_2 \left( D_{z,\alpha}^2, D_{w,\beta}^2 \right) = 0.
$$

The other cases are proved in a straight way.