[Reviews · NeurIPS 2018]

Reviewer 1



This paper introduces a new metric for quantifying the distance or difference between two dynamical systems. The metric is a generalization of the principle angle metric, Martin's metric, and the Binet-Cauchy metric. The primary innovation and fundamental insight of this paper is to formulate the metric as a ratio of reproducing kernels calculated on the orbits of two dynamical systems. The idea is innovative indeed. My suggestions to this paper are minor. First, the authors ought to prove that the so-called ratio reproducing kernel Hilbert space (RKHS) metric satisfies the properties of a classical metric function, namely positivity (I believe Proposition 3.1 addresses this), symmetry, and the triangle inequality. This is a minor clarification that would be easy to add. Second, I would encourage the authors to improve the readability of the paper. The overall flow of the paper is stymied by halting exposition and long strings of unannotated mathematical displays. I struggled to find the definition of the metric (the main point of the paper) on lines 89-93. There were three distinct mathematical objects defined in that short span of space, but none of them are formally named or introduced. The authors ought to emphasize the utility of their findings. They clearly show their ratio "metric" generalizes prior metrics, but beyond that, it should be clarified what the practical utility is for this generalized metric. Overall, the technical content of this paper is interesting and should be accepted eventually.

Reviewer 2



Summary: The paper “Metric on Nonlinear Dynamical Systems with Koopman Operators” proposes a general metric for nonlinear dynamic systems using the Koopman operator useful for learning with structural data and an approximation for finite data. The general metric is shown to contain as special cases established metrics for ARMA models and dynamic systems such as Martin’s and Binet-Cauchy metric. This contribution seems to extend the papers “Koopman spectral kernels for comparing complex dynamics: Application to multiagent sport plays” and “Dynamic Mode Decomposition with Reproducing Kernels for Koopman Spectral Analysis” while also unifying related metrics from Vishwanathan et al. and Martin. The metric shows good discriminating behavior used in Szegö and Guassian kernel demonstrated on synthetic rotation dynamics on the unit disk and 3 real world one-dimensional time series taken from sensor and device data. Qualitative Assessment: Since metrics for kernelized analysis of dynamic system trajectories gains increasing interest in the community this contribution is generally interesting. There is no comparison between alternative ML methods for structural data such as HMM or recurrent neural networks, but I see that this paper is meant mainly as theoretical basis and not a thorough investigation of its inclusion in ML algorithms yet as indicated in the outlook. I found the paper extremely hard to read. There are a lot of spaces where some simple sentences accompanying the mathematical notion might help. For example this is the first time I see the logical and operator ^ in context with a trace operation and I am not sure what it exactly means … probably it is denoting the subspaces together with m which is just introduced as a natural number in line 89. If the paper is written in a more accessible way I can imagine it can gain a lot of citations. Furthermore, it would be useful for practical application in the experimental section to mention the sampling of the finite data in the experiments and on the rotation example one could also compare the influence of the sampling density on the metric quality. References: - [21] Williams et al. “A data-driven approximation of the Koopman operator: Extending dynamic mode decomposition.” is mentioned in the experimental section wrt the data which is used, but it is not discussed how it relates to this work. - missing: Steven et al. “Koopman invariant subspaces and finite linear representations of nonlinear dynamical systems for control,” 2015 - Kernel Mean Embedding might be interesting as well: A. Smola, A. Gretton, L. Song, and B. Schölkopf, in Algorithmic Learning Theory: 18th International Conference, 2007 Minor comments: - Line 179 (C 2 .A 2 , I q ) → . instead of , - Line 188 the trace kernel k_tr seems to miss the trace - Line 264 means Figure 3 instead of 2 - Line 269 … discriminated compared → discriminated better compared The authors addressed the points raised by the reviewers, promised to improve the readability and add experimental investigation of the proposed metric on examples.

Reviewer 3



* Summary : This paper introduces a general framework to compute a metric between deterministic dynamical systems. This framework relies on the Koopman operator computed in RKHS, which was previously introduced in [Dynamic mode decomposition with reproducing kernels for koopman spectral analysis, Kawahara, NIPS18]. The framework is generic and the authors show that it encompasses previous metrics introduced for linear dynamical systems [Subspace angles between ARMA models, De Cock and De Moor, Systems and Controls Letter 2002] and non-linear ones [Binet-Cauchy kernels on dynamical systems and its application to the analysis of dynamical scenes, Vishwanathan et al.. 2007]. An algorithm is proposed in order to compute this metric from the observation of multiple trajectories generated by these different dynamical systems. Visualization experiments on synthetic and real data show that this metric seems to separate dynamical systems with different properties. * Review : This paper is interesting and brings a new framework to study dynamical systems with the Koopman operator, which makes it a significant contribution. Clarity should be improved. I think that the notations and objects introduced need to be greatly slimmed down to simplify the exposition, just keeping the minimum to reach the main goals of the paper. Further, it would be nice to provide more intuition on the different objects which are introduced in Subsection 3.1. When it comes to experiments, the visualizations provided are nice, but running a true machine learning algorithm would have been better. For instance, on the real time series, it would have been nice to run a clustering algorithm on top of the metrics computed, to see for instance if it is able to retrieve the original classes, yielding more quantitative measurements. Overall, I would tend in favor of accepting this paper, provided it is clarified. * Detailed remarks: - As far as I understand, $\tilde{A}_m$ is introduced to allow convergence even in case of admissibility and not semi-stability (cf Prop 3.3). This is a nice results, but it is not exploited in the experiments. Why, or why not? - In Sec 3.1, it would be nice to provide more intuition on the different objects h, L_h, and the subspaces H', H''. What do they represent? Maybe notations could be simplified to introduce only L_h and remove h? - Is the object $d_m^T$ a positive definite kernel? Or a metric? I have not seen it stated anywhere, or any hint of proof of it (I am not sure it is even named). Since this object is very crucial, it may deserve a definition in its own right and much more explanations or intuitions. What is the role of the parameter $m$ for instance? - What was the motivation for introducing $A_m^T$ from $d_m^T$? Is it only to take into account all previous works in this framework, or is there something more subtle at work? Is it obvious that $A_m^T$ defines a positive definite kernel? - Indices and notations need to be slimmed to the minimum over all the paper. For instance, I am not sure that Proposition 2.1 brings much to the main goal of the paper, as it is not used anywhere else. Using multiple $A$ in 4.1 brings confusion.